# Immunoreactive Trypsinogen in Infants Born to Women with Cystic Fibrosis Taking Elexacaftor–Tezacaftor–Ivacaftor

**DOI:** 10.3390/ijns9010010

**Published:** 2023-02-21

**Authors:** Payal Patel, Jana Yeley, Cynthia Brown, Melissa Wesson, Barbara G. Lesko, James E. Slaven, James F. Chmiel, Raksha Jain, Don B. Sanders

**Affiliations:** 1Indiana University School of Medicine, Indianapolis, IN 46202, USA; 2Department of Medicine, Indiana University School of Medicine, Indianapolis, IN 46202, USA; 3Department of Molecular Genetics, Indiana University School of Medicine, Indianapolis, IN 46202, USA; 4Department of Pathology, Indiana University School of Medicine, Indianapolis, IN 46202, USA; 5Department of Biostatistics and Health Data Science, Indiana University School of Medicine, Indianapolis, IN 46202, USA; 6Department of Pediatrics, Indiana University School of Medicine, Indianapolis, IN 46202, USA; 7Department of Internal Medicine, University of Texas Southwestern Medical Center, Dallas, TX 75390, USA

**Keywords:** cystic fibrosis, newborn screening, immunoreactive trypsinogen

## Abstract

Most people with cystic fibrosis (CF) are diagnosed following abnormal newborn screening (NBS), which begins with measurement of immunoreactive trypsinogen (IRT) values. A case report found low concentrations of IRT in an infant with CF exposed to the CF transmembrane conductance regulator (CFTR) modulator, elexacaftor–tezacaftor–ivacaftor (ETI), in utero. However, IRT values in infants born to mothers taking ETI have not been systematically assessed. We hypothesized that ETI-exposed infants have lower IRT values than newborns with CF, CFTR-related metabolic syndrome/CF screen positive, inconclusive diagnosis (CRMS/CFSPID), or CF carriers. IRT values were collected from infants born in Indiana between 1 January 2020, and 2 June 2022, with ≥1 CFTR mutation. IRT values were compared to infants born to mothers with CF taking ETI followed at our institution. Compared to infants identified with CF (*n* = 51), CRMS/CFSPID (*n* = 21), and CF carriers (*n* = 489), ETI-exposed infants (*n* = 19) had lower IRT values (*p* < 0.001). Infants with normal NBS results for CF had similar median (interquartile range) IRT values, 22.5 (16.8, 30.6) ng/mL, as ETI-exposed infants, 18.9 (15.2, 26.5). IRT values from ETI-exposed infants were lower than for infants with abnormal NBS for CF. We recommend that NBS programs consider performing CFTR variant analysis for all ETI-exposed infants.

## 1. Introduction

Cystic Fibrosis (CF) is a rare autosomal recessive disease caused by mutations in the CF Transmembrane Conductance Regulator (CFTR) gene, which encodes an epithelial chloride and bicarbonate transport channel [1]. CFTR modulators, small-molecule oral medications that improve CFTR function, have substantially improved the lives of people with CF [2]. Elexacaftor–tezacaftor–ivacaftor (ETI) was approved by the US Food and Drug Administration (FDA) for adolescents and adults with CF and ≥1 copy of the F508del variant in the CFTR gene at the end of 2019. Some people with CF with CFTR variants other than F508del have also gained access to ETI as a result of positive in vitro testing results [3]. Because of the unknown effects of ETI on fetal development, pregnant women were not included in clinical trials of ETI [4,5]. Following FDA approval, studies of women with CF taking ETI during their pregnancy have demonstrated greater or similar concentrations of ETI components in umbilical cord blood compared to maternal serum concentrations, as well as low amounts of all ETI components in breast milk [6]. There are no reports of in utero exposure to ETI in animal models, but in utero exposure to ivacaftor in G551D homozygous ferrets protected against the development of a CF phenotype (including pancreatic insufficiency), which was maintained during post-natal ivacaftor administration [7].

Most people with CF are now diagnosed following an abnormal newborn screening (NBS) test, which begins with the measurement of immunoreactive trypsinogen (IRT), which is typically elevated in newborns with CF [8]. Depending on the local NBS protocol, those with high concentrations of IRT may have a second measurement of IRT, a measurement of pancreatitis associated protein, and/or a test for one or more CFTR variants.

Previously, some women with CF experienced fertility issues and/or were hesitant of becoming pregnant due to the risk of complications [9]. ETI has led to improved health and likely an increase in fertility amongst women with CF, increasing their desire and ability to have children [10,11]. There has been a case report of a low concentration of IRT in an infant with CF exposed to ETI in utero leading to false normal NBS results [12]. An overall assessment of IRT in infants born to mothers taking ETI has not been conducted. We hypothesized that infants born to mothers with CF taking ETI (ETI-exposed) while pregnant may have lower concentrations of IRT than newborns with CF, CFTR-related metabolic syndrome/CF screen positive, inconclusive diagnosis (CRMS/CFSPID), or CF carriers. Since ETI-exposed infants are obligate carriers, if they have lower concentrations of IRT, then the risk of false normal NBS results would be increased.

## 2. Materials and Methods

NBS in the State of Indiana begins with the measurement of IRT from a single dried blood spot. Infants with IRT concentrations in the top 4% for the day have CFTR variant analysis completed on the same dried blood spot. The CFTR variant panel screens for 39 common variants. Infants are referred for sweat testing if one or more CFTR variants are identified. Infants with IRT above 170 ng/mL without any identified CFTR variants are not automatically referred for sweat testing in Indiana.

For this study, NBS data were collected from all infants born in Indiana between 1 January 2020, and 2 June 2022, with at least one CFTR variant. IRT concentrations were compared to NBS data of infants born in the same period to mothers with CF taking ETI and followed at the Indiana University CF Care Center. All of the mothers continued ETI throughout their pregnancies. Infants were categorized into the following groups: CF, CRMS/CFSPID, Carrier, Unknown (i.e., a diagnosis was not determined), and ETI-exposed. Descriptive statistics were used to summarize IRT concentrations by group; we report mean and median values as the data was not normally distributed. Analysis of Variance models (ANOVAs) were created using log-transformed data with Dunnett-adjusted *p*-values to compare IRT values between the reference ETI-exposed group and the other categories [13]. Analyses were performed using PRISM 9.4 (GraphPad Software, San Diego, CA, USA) and SAS v9.4 (SAS Institute, Cary, NC, USA). This project was approved on 2 June 2022 by the Indiana University IRB (Protocol #15375).

## 3. Results

Between 1 January 2020, and 2 June 2022, there were 190,493 infants born in Indiana who underwent NBS, including 636 infants who had an elevated IRT and ≥1 identified CFTR variant. Of these, 51 were diagnosed with CF, 21 with CRMS/CFSPID, and 489 as CF carriers (Table 1). The incidence of CF was 1 in 3735 live births. Two-thirds of infants with ≥1 identified CFTR variant had ≥1 copy of F508del; this likely owes to the study design and the racial and ethnic makeup of the screened population in the state of Indiana. Mean and median IRT values were greatest for the infants diagnosed with CF and least for CF carriers. There were 75 infants without a confirmed diagnosis; most often this was because sweat testing was not completed successfully, or the child was lost to follow up or died. The unknown diagnosis group had IRT concentrations similar to CF carriers.

During the same time period, there were 19 infants born to mothers taking ETI and followed at the Indiana University CF Care Center. To our knowledge, none of these infants has CF. Only three of the 19 infants had CFTR variant analysis completed; two infants were tested because of the presence of respiratory symptoms and one was tested upon parental request. One infant had 0 variants detected on the 39-variant NBS panel (consistent with the maternal genotype), and two infants were identified as CF carriers. Compared to the infants with CF, CRMS/CFSPID, CF carriers, or unknown diagnosis, ETI-exposed infants had significantly lower mean and median IRTs (*p* < 0.001 for each comparison) (Table 1 and Figure 1).

During the same time period, 189,857 infants born in Indiana underwent NBS for CF and had no identified CFTR variants, i.e., infants with an IRT not in the top 4% for the day, or in the top 4% for the day but with no CFTR variants identified, and were not diagnosed with CF through other means, e.g., meconium ileus. None of the infants with IRT not in the top 4% for the day had CFTR variant analysis because of their low IRT values. These infants had similar median IRT values to the ETI-exposed infants, *p* = 0.41 (Table 1).

## 4. Discussion

Women taking ETI during pregnancy, and their partners, will be understandably eager to have their newborn infants appropriately diagnosed with CF, CRMS/CFSPID, or as CF carriers. Building on case reports of false normal NBS for CF in infants born to women being treated with ETI [12,14], we describe IRT values for infants with in utero exposure to ETI. ETI-exposed infants had similar IRT concentrations as infants with normal NBS results, and lower IRT concentrations than expected for CF carriers [15]. To our knowledge, no ETI-exposed infants in Indiana have been diagnosed with CF, though CFTR variant analysis has not been performed for most of these infants.

Having IRT concentrations below cutoff values is the most common cause of false negative NBS for CF [16,17]. Therefore, it was unclear whether the published case report of a false negative NBS for CF in an ETI-exposed infant was directly related to the ETI exposure. The low IRT levels demonstrated in infants exposed to ETI in this report raises the possibility of false normal NBS and delayed, or missed, diagnoses. Notably, the IRT concentrations among ETI-exposed infants were uniformly low without any outliers; to our interpretation, this may support a true effect of ETI on IRT. Indiana uses a floating IRT cutoff value, which is preferred (compared to a fixed cutoff) [18,19], so it is unlikely that these results are primarily due to the NBS algorithm.

Testing of the partner prior to pregnancy may not entirely rule out CF in the infant, depending on the CFTR variant panel and ethnic background of the partner [20]. It is also unknown to what extent ETI exposure through breastfeeding could interfere with sweat test results. In the case report noted above [12], the ETI-exposed infant with CF (homozygous F508del) had sweat chloride values at 5 weeks of age of 60 and 67 mmol/L, which are lower than expected for an infant who is homozygous for the F508del mutation, and indicates that sweat chloride values in ETI-exposed infants may not reliably identify infants with CF. Therefore, it is possible that the parents of an ETI-exposed infant may be falsely reassured without an assessment for CFTR variants in the infant. Furthermore, the CFTR variant panel used for NBS may not include the mother’s CFTR variants, which occurred for one of the ETI-exposed infants born in Indiana. Even though the likelihood of a missed CF diagnosis is low, there are policy implications for correctly identifying infants with CRMS/CFSPID or even CF carriers that need to be considered [21]. To our knowledge, there are no other NBS programs that consider maternal medications in the interpretation of the results of NBS testing.

Limitations of our study include having a small sample size, the possibility that ETI-exposed infants were not reported to the Indiana University CF Care Center or are followed elsewhere, and no assessment of newborn-related factors that may affect IRT (e.g., age, premature birth, race) [17]. Race was not reliably tracked in NBS records. This analysis grouped infants based on current CF-related diagnosis and does not account for infants later diagnosed with CF after negative NBS or the large number of CF carriers born in Indiana who were not identified with an elevated IRT. Indiana’s NBS program identified approximately 10 CF carriers for every CF diagnosis, and approximately 2.5 CF diagnosis per CRMS/CFSPID diagnosis, both of which are consistent with reports from other US states’ NBS programs [22]. Given the lack of systematic follow up, we did not include information on sweat test results, subsequent CFTR testing, or children diagnosed with CF through prenatal screening or who presented symptomatically.

## 5. Conclusions

The mean concentration of IRT in ETI-exposed infants was lower than for infants with abnormal NBS for CF, supporting the concern of false normal NBS results. Given the importance of early diagnosis for infants with CF [23], and to minimize parental anxiety, we recommend that NBS programs consider performing CFTR variant analysis on all ETI-exposed infants regardless of the IRT concentration. CFTR variant analysis should be informed by the parents’ known CFTR variants and race and ethnicity [24]. Performing CFTR variant analysis for all ETI-exposed infants would allow parents to be informed of NBS results that are not diagnostic for CF, which is the option parents prefer [25]. In order for CFTR variant analysis to be performed for all ETI-exposed infants, close communication between the Adult CF Care Center, the state NBS laboratory, and the Pediatric CF Care Center is necessary to allow for timely analysis of CFTR variants from the already-collected dried bloodspot.

## Figures and Tables

**Figure 1 IJNS-09-00010-f001:**
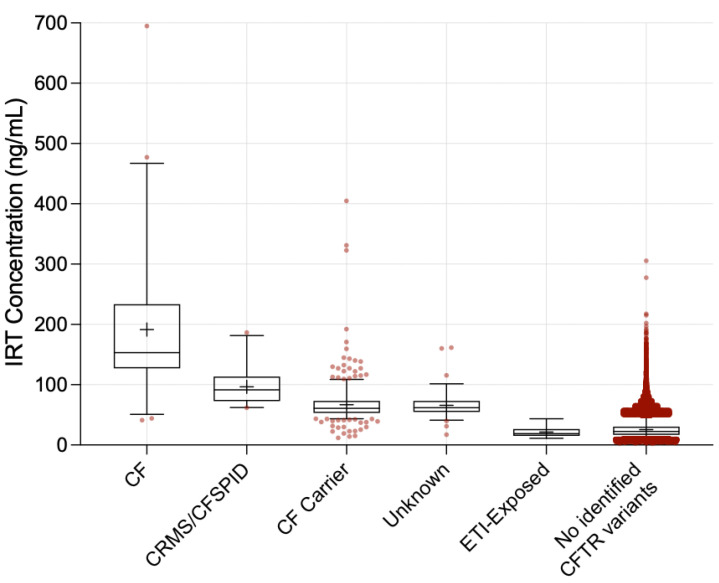
Box-and-Whisker Scatter Plot of IRT Concentrations (ng/mL). This figure shows the distribution of IRT values from NBS in each infant group. Boxes represent the 25th–75th values; the horizontal line within each box indicates the median value; the cross within the box represents the mean. Whiskers indicate the 5th to 95th percentiles. Each dot represents the results for an individual infant that fell outside the 5th and 95th percentiles.

**Table 1 IJNS-09-00010-t001:** Cohort characteristics. Infants born in Indiana between 1/1/20 and 6/2/2022 with ≥1 Cystic Fibrosis (CF) Transmembrane Conductance Regulator (CFTR) variant, in utero exposure to elexacaftor–tezacaftor–ivacaftor (ETI), or normal immunoreactive trypsinogen (IRT) values.

Infant Group	*N*	≥1 F508del (%)	Mean IRT (ng/mL)	Median IRT (ng/mL)	IQR	*p*-Value ^3^
CF ^1^	51	98.0%	191.6	153.1	127.1–233.6	<0.001
CRMS/CFSPID ^1^	21	80.9%	96.7	91.5	73.0–112.2	<0.001
CF Carrier ^1^	489	62.8%	66.9	60.8	53.1–73.2	<0.001
Unknown Diagnosis ^1^	75	65.3%	65.7	61.8	54.8–73.1	<0.001
ETI-Exposed	19	n/a ^2^	21.6	18.9	15.2–26.5	-
No identified CFTR variants	189,857	0.0%	25.4	22.5	16.8–30.6	0.41

Abbreviations: Cystic fibrosis, CF; CF Transmembrane conductance Regulator, CFTR; CFTR-related metabolic syndrome/CF screen positive, inconclusive diagnosis, CRMS/CFSPID; elexacaftor–tezacaftor–ivacaftor, ETI; immunoreactive trypsinogen, IRT; interquartile range, IQR; newborn screening, NBS. ^1^ With ≥1 CFTR variant on NBS. ^2^ CFTR variants have only been analyzed for 3 infants; only 1 had a copy of F508del identified. ^3^ Compared to ETI-exposed group.

## Data Availability

The information collected from the newborn screening program was obtained under conditions that the data would not be made publicly available.

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
