# Peer review of "Immunoreactive Trypsinogen in Infants Born to Women with Cystic Fibrosis Taking Elexacaftor–Tezacaftor–Ivacaftor"

_2409-515X, 2023, doi:10.3390/ijns9010010_

Round 1
Reviewer 1 Report
This manuscript describes a systematic investigation of immunoreactive trypsinogen (IRT) values in infants born in Indiana between January 1, 2020 and June 2, 2022, with specific focus on values in infants with CF, CRMS, CF carriers, unknown diagnoses, infants with normal IRT values (less than the 96th percentile), and importantly, IRT values in infants whose mothers were receiving elexacaftor-tezacaftor-ivacaftor (ETI).
Major comments
1. Kudos to the authors for using the term “false normal” newborn screening results (instead of the term false negative).
2. Page 3, lines 102-104 and page 4, lines 157-158: There were 3 infants born to mothers with CF receiving ETI who had CFTR variant analysis completed. Two infants were identified as carriers and one infant had no variants detected on the 39-variant newborn screening panel (consistent with the mother’s genotype). Obviously, the infant who had no variants detected actually has a variant as he/she must be, at a minimum, a carrier. This fact may not be obvious to readers who do not know that CF is an autosomal recessive disease. I agree that CFTR variant analysis should be performed on infants of mothers with CF who are receiving ETI. The discussion can be strengthened by stating that the variant analysis should include the mother’s CFTR variants.
3. Reference number 9 are case reports of infants born to mothers who are receiving ETI. In this reference, there was one infant who had CF. The infant’s IRT was less than the 95th percentile. At the request of the CF center, variant analysis was performed and demonstrated that this infant was homozygous F508del. At 5 weeks of age, this infant had sweat chloride values of 60 and 67 mmol/L. This is significantly less than expected for a patient who is homozygous F508del. In the discussion, the authors can strengthen their recommendation of proceeding directly to variant analysis in infants of mothers receiving ETI (instead of starting with a sweat test) by citing that sweat chloride values in infants of mothers who are receiving ETI may not reliably discriminate CF from normal.
Reviewer 2 Report
Immunoreactive trypsinogen in infants born to women with CF taking elexacafor-tezacaftor-ivacaftor (ETI).
Brief summary
The authors have undertaken a study in the State of Indiana, on initial IRT measurement in newborn screened infants (NBS) for CF born between 2020 and mid-2022. They focused on those who carry at least one CFTR variant including the obligate carriers ETI - exposed infants. Their hypothesis was that infants born to mothers with CF taking ETI during pregnancy may have lower IRT values, raising the risk of a higher rate of false normal NBS results compared to newborns with CF, CRMS/CFSPID, or carriers. The results are in favor of their hypothesis, thus they recommend that all ETI-exposed infants have a CFTR variant analysis performed.
The manuscript is very clear, very well written, presented in a well-structured manner, and relevant to the field. This is the first study in the field, literature is mainly from case reports.
The design of this observational retrospective study is appropriate for testing the hypothesis.
The ethical committee approved the study.
Statistical analysis is not part of my expertise but appears correctly done. The table and Figure properly show the data. The discussion is comprehensive.
Conclusions are consistent with the data presented.
References are missing [20][21]and [22].
Comments
Abstract
Line 28 “We recommend that all ETI-exposed infants have a CFTR variant analysis performed”. In my opinion, this statement may be too strong: 1. is based on an expert opinion; 2. The study population was limited to 19 infants born to mothers with CF taking ETI during pregnancy. It should be revised.
Results
Line 102-104. You mention that “only three of the 19 infants ETI-exposed have CFTR variant analysis completed”. Could you provide the reason for this analysis: were these infants symptomatic? Was it related to parental anxiety? Was it related to the recent publication of an FN case? Was it related to CF doctor anxiety?
Table. For improving the reading of the table, I suggest drawing a line between Unknown diagnosis and ETI-exposed and a line between ETI-exposed and Normal-IRT.
Line 111: it is written Table1 but line 95 is written Table
Figure 1: the plots for Normal-IRT show a huge variety of values in the lower range (<96 centiles) but also in the very high range. The box plot Normal-IRT concerns infants with IRT <96 percentile plus those with IRT>170 who have no detected variant of the 39]. Am I correct? If it is the case maybe this can be explained in the Figure comments at the bottom or in the text. They are not really all with Normal –IRT. Or an option may be to show only those with IRT below the cut-off.
Discussion
Line 118 “People taking ETI during pregnancy “can be changed to “Women taking ETI during pregnancy “
Line 132: “to our interpretation, this supports a true effect of ETI on IRT” can be changed to “to our interpretation, this may support a true effect of ETI on IRT”. The cohort is limited thus replication of a study or extension of the study is warranted.
Conclusion
Lines 156-165: the conclusion may improve in clarity by moving the order of the sentences.
First lines 155-156, then a sentence on the recommendation of information for women who use highly effective modulators during pregnancy that may result in a false negative screen infant. Then sentences 156-161 may be reformulated. And lastly a sentence on communication “Therefore, for ETI-exposed infants if the father’s genotype is unknown, or if the father is known to be a carrier, the child should undergo genotyping following a negative NBS test from the already collected blood.”
References
[4] Sun X pages are missing
[8] Jain R pages are missing
[14] Kharrazi M pages are missing]
[17] Mc Garry ME pages are missing
[20] Reference is missing
[21] Reference is missing
[22] Reference is missing
Reviewer 3 Report
This study describes the effect of CFTR modulators (ETI) on IRT as the primary marker in the algorithm for the cystic fibrosis (CF) newborn screening. ETI during pregnancy and their effects in utero and on the newborn is an important topic, as the number of pregnancy in women with CF has been increasing since FDA approval for ETI in patients with >= 1 copy of the F508del variant at the end of 2019.
The result of this study is that 19 ETI-exposed infants had lower IRT values compared to infants identified with CF, CFSPID and CF carriers. Although this is an interesting finding, it would be much more interesting to know the outcome of the infants in general, as there are only a few infants born so far to mothers taking ETI and pregnant women were not included in clinical trials of ETI.
I think it is standard anyway to perform a CFTR variant analysis in infants born to mothers with CF and not to rely on IRT values that are known to be sometimes falsely negative. This recommendation is also the conclusion of the authors, so it is astonishing that only in three of their 19 infants a CFTR variant analysis was performed?!
Minor issues:
A sentence about cystic fibrosis and CFTR modulator therapy in general would be nice at the beginning of the introduction.
Table ("1" is missing):
The abbreviations should be explained.
The 5.3% in the group exposed to ETI is misleading as this is only one child, but out of three children that have been analyzed for CFTR variants. I would delete this value and include it in the footnote.
Normal-IRT should read "normal IRT or no CFTR mutation" as there must be a group with elevated IRT values but none of the 39 CFTR mutations. The same is true for the text and Figure 1 (and 300ng/ml is certainly not a normal IRT value)
Figure 1:
The design/color should be changed as boxes and mean are not visible in some columns.
The column "Normal-IRT" seems to be wrong as the IQR in Table 1 is 16.8-30.6?
Lines 102-103: ETI is approved for CF patients with at least one copy of the F508del variant in the CFTR gene. How is it possible that a child and mother with none of the 39 CFTR mutations are in the study group?
References 20-22 are missing.
Authors' contributions: lines 167-168 and 175-177 should be deleted.
Round 2
Reviewer 2 Report
I validate the authors' new manuscript. They made the changes required. Just one modification should be done : At the bottom of the Table, Abbreviation : CF screen positive indeterminate diagnosis is incorrect and should be CF screen positive, inconclusive diagnosis.
Author Response
Thank you for catching this mistake. We have corrected it in the Abstract, Methods, and the Table.
Reviewer 3 Report
The paper is much better now
Author Response
Thank you for the thoughtful review.